# Hydrothermal Surface Treatment of Biodegradable Mg-Materials

**Andrzej Miklaszewski** **, Kamil Kowalski** **and Mieczyslaw Jurczyk \***

Institute of Materials Science and Engineering, Poznan University of Technology, Jana Pawla II 24,
61-138 Poznan, Poland; andrzej.miklaszewski@put.poznan.pl (A.M.); kamil.kowalski@put.poznan.pl (K.K.)
**\*** Correspondence: mieczyslaw.jurczyk@put.poznan.pl; Tel.: +48-61-665-3508

**Abstract:** Paper presents study on the hydrothermal treatment for hydroxyapatite layer formation on the different biodegradable Mg-substrates. The evaluation of corrosion resistance in Ringer's solution and contact angle measurements in glycerol were performed. Alloys and composites substrates obtained by mechanical alloying and powder metallurgy route are characterized by submicron range microstructure, which is responsible for further surface processing characteristic. Hydrothermal treatment in Ca-EDTA (ethylenediaminetetraacetic acid calcium disodium salt) led to formation of hydroxyapatite layers, which improves both the corrosion resistance and surface wetting properties compared to microcrystalline magnesium.

**Keywords:** magnesium; mechanical alloying; hydrothermal treatment; hydroxyapatite coating; contact angle measurements; corrosion resistance

---

## 1. Introduction

Magnesium alloys and their composites strongly attract the international community of researchers because of their potential as temporary biodegradable implant materials, mostly appreciated for their physical and mechanical properties [1]. In spite of many advantages of these materials, the weaknesses associated with the influence of the surrounding tissues and human body fluids remain the most concerning ones related to mechanical integrity, pH increase, or hydrogen evolution. To effectively reduce fast degradation rates and enhance corrosion resistance characteristics, different bulk or surface approaches could be proposed [2]. For the surface magnesium modification, the researchers' attention focuses on the method with a great potential of application. The treatment techniques such as electrophoretic [3], microwave-assisted [4], plasma or laser deposition [5], hydrothermal [6,7] or other [8–11] showing promising results remain in the scope of interest as a method that is environmentally friendly and amenable to scale up. Crucial from the practical point of view, anticorrosion coatings, obtained in the environment-friendly, single-step process, stay as the most desirable products, especially when the materials like magnesium are considered. Low corrosion resistance of the magnesium in different aqueous solutions arise from its high ionization tendency, lead to the reactions in a wet environment at pH below 11 [12]. The potential growth that activates the release of the Mg ions may in some cases prevent the formation of protective layers, as also its crystallisation.

Hydroxyapatite (HAp) layers obtained by the hydrothermal treatment approach proposed in this work may offer high corrosion resistance and no environmental toxicity. Additionally, they are characterized also with a high thermodynamic, structural stability, and, most importantly, basic components that are the same as human bone. High Mg ions concentration in aqueous solutions is problematic but not impossible to overcome at the layer formation stage, and it is solved in the method by a Ca chelate compound, which can maintain a sufficiently high concentration of Ca ions that allows

to synthesise a HAp layer on magnesium substrates. Proper treatment temperature and concentration of calcium and phosphate ions improve the protectiveness of HAp coatings [6].

For bulk approach, we may distinguish compositional changes realized through a development of new alloys and composite structures [13]. Additionally, the microstructure size control and phases distribution play a crucial role in the enhancement of the properties. The above-mentioned structural relation factors influence the volumetric and surface material response through interactions among different phases. Alloying remains as one of the most effective methods of improving the corrosion resistance and mechanical properties of magnesium [14]. Zinc, for example, as an element highly essential for humans, helps overcome the harmful corrosive effect of iron and nickel impurities. Manganese improves the salt-water resistance of the magnesium alloy by removing iron and other heavy-metal elements [15]. The zirconium containing the magnesium alloys, besides improving the mechanical properties, usually has a higher corrosion resistance compared to the Zr-free magnesium materials [16]. Other elements such as silver, for example, could be considered as antibacterial surface agents [17], also widely investigated in titanium-based composite systems [18]. Several investigations have been done to study the effect of rare earth elements when they are alloyed with pure magnesium and other Mg alloys [19,20]. For example, different studies examine the microstructure, mechanical, and corrosion-related properties as well as in vitro cytocompatibility of the bulk ultrafine-grained Mg-4Y-5.5Dy-0.5Zr material through 45S5 Bioglass alloying [21]. The ultrafine grained (Mg-4Y-5.5Dy-0.5Zr)-5 wt.% 45S5 Bioglass composite exhibits higher corrosion resistant properties than the bulk Mg-4Y-5.5Dy-0.5Zr alloy after the HF (Hydrofluoric acid) treatment. The in vitro biocompatibility of the synthesized composites was evaluated and compared with microcrystalline magnesium. Magnesium light-weight matrix composites may offer high attractiveness for different commercial products and application when proper processing route is considered [22]. The magnesium composites based on Mg-4Y-5.5Dy-0.5Zr alloy with 5 wt.% 45S5 Bioglass or 5 wt.% 45S5 Bioglass and 1 wt.% Ag addition modified with $MgF_2$ exhibit better biocompatibility in comparison to base Mg material. Additionally, different surface modification treatments could be proposed, attractive for their possible improvements of the corrosion resistance and biocompatibility of the magnesium-based materials [23,24]. Based on the previous research, bulk and surface treatment relations were applied and combined in this paper to maximize the effect of property enhancement obtained independently from the composition, the structure, as well as the surface treatment procedure. The samples of the proposed alloys and their composite structure obtained through mechanical alloying (MA) and the proposed surface hydrothermal treatment (HT) procedure discussed in this paper, significantly outrun a typical material improvement approach. The proposed conceptualization leads to a broader scope in the direct enhancement of the properties of biodegradable Mg-materials and, most importantly, at the same time allows an analysis of the influence of different factors simultaneously (the influence of composition or surface condition on the corrosion resistance and contact angles measurements).

## 2. Materials and Methods

### 2.1. Sample Preparation

The magnesium alloys and the bioceramic composites based on these alloys were prepared by MA and powder metallurgy where bioceramic = hydroxyapatite (HAp) and 45S5-Bioglass (BG) according to Figure 1 processing scheme. The proposed weight proportions were also modified by the addition of silver as follows:

1. Mg pure—Reference sample
2. Mg4Y5.5Dy0.5Zr
3. Mg4Y5.5Dy0.5Zr + 5%BG
4. Mg4Y5.5Dy0.5Zr + 5%BG + 1%Ag
5. Mg1Zn1Mn0.3Zr

6.   Mg1Zn1Mn0.3Zr + 5%HA
7.   Mg1Zn1Mn0.3Zr + 5%HA + 1%Ag
8.   Mg1Zn1Mn0.3Zr + 10%HA

Elemental magnesium (99.8%, 45 μm; Alfa-Aesar, Karlsruhe, Germany), zinc (99%, 600 μm; Alfa-Aesar), manganese (99%, 45 μm; Alfa-Aesar), yttrium, dysprosium (99.9%; approx. 100 μm, Alfa-Aesar), zirconium (95.0%, 350 μm; Ciech, Inowroclaw, Poland), hydroxyapatite (reagent grade; Sigma-Aldrich, Darmstadt, Germany), and amorphous 45S5 Bioglass (53 μm, Mo-Sci Health Care L.L.C., Rolla, MO, USA) powders were used as starting materials. The powder mixtures were weighed according to the weight percentage compositions, blended and processed by MA, similarly to our previous work [19]. Mechanical alloying was performed using SPEX 8000 Mixer Mill (SPEX SamplePrep, Metuchen, NJ, USA) for 48 h, where elemental powders were weighed, blended and poured into vials in a glove box (Labmaster 130, MBraun, Garching, Germany) filled with automatically controlled inert argon atmosphere ($O_2 <$ 2 ppm and $H_2O < 1$ ppm). A weight ratio of hard steel balls (10 mm diameter) to powder equaled 10:1. The obtained powders were then cold compacted by uniaxial mode with the pressure of 0.6 GPa into the pellets (ϕ8 and 3 mm height). Then, the green compacts were sintered at 550 °C for 2 h in the argon atmosphere (99.999% purity) to form the bulk samples.

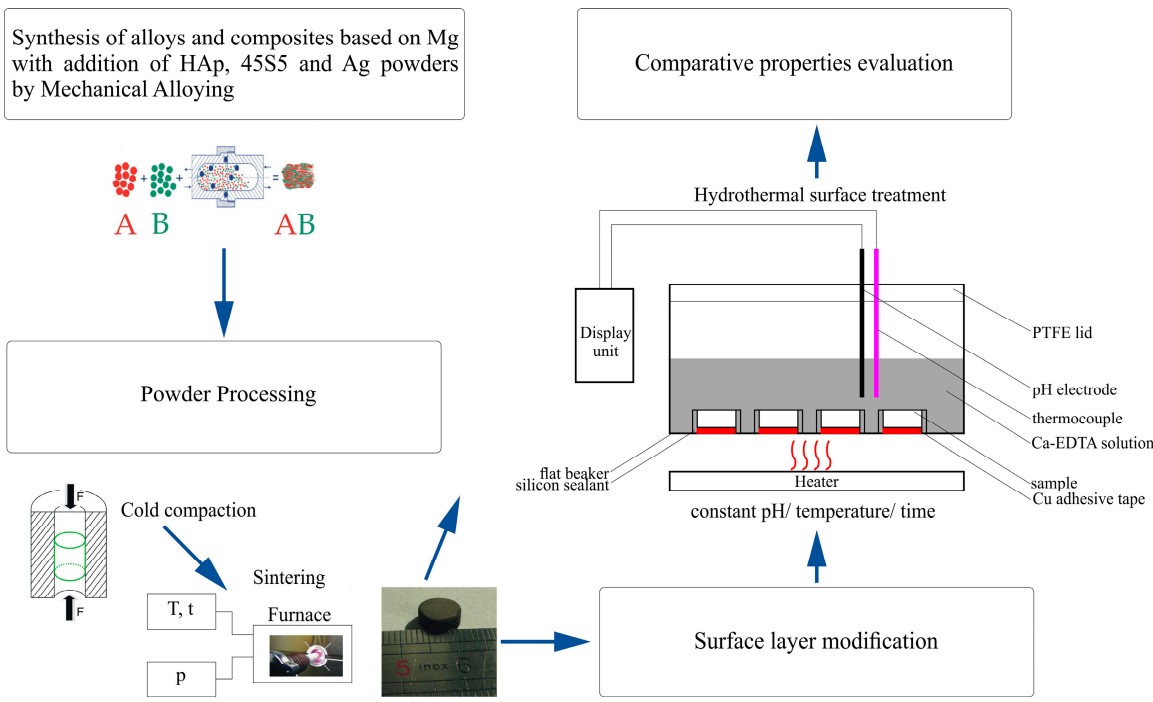

**Figure 1.** Schematic illustration of the experiment arrangement.

## 2.2. Surface Hydrothermal Treatment

For the surface hydrothermal treatment, the prepared sintered samples were grinded with sand paper (400 grit) and rinsed ultrasonically in acetone. Then, the samples were placed on the bottom of the beaker and isolated by an adhesive silicon sealant in the way that enables the contact with the electrolyte wet environment only on one base side of the sample. Before HT, the samples were chemically polished with 8 vol.% $HNO_3$-1 vol.% $H_2SO_4$ for 10 s and flushed with distilled water to remove the impurities. The prepared samples were then treated hydrothermally at $100 \pm 0.2$ °C in a fresh 0.25 mol/L Ca-EDTA (Sigma-Aldrich) and 0.25 mol/L $KH_2PO_4$ (Sigma-Aldrich) solution whose pH was earlier adjusted by a 1 mol/L NaOH to the value of 8.9. The proposed concentration of the solution and time (7.2 ks) were based on other research [6,25,26] in order to prune the treatment time necessary for the formation of the hydroxyapatite layer.

## 2.3. Structural and Morphological Surface Analysis

The hydrothermally treated surfaces were analyzed after sample drying and a 24 h desiccator storage. The X-ray diffraction (XRD) analysis was conducted on the Panalytical Empyrean equipment with the copper anode (CuK$_\alpha$—1.54 Å). The measurement conditions for all the collected data were set up for 25–60° with a step size of 0.015°/15 s and 45 kV/40 mA. For the obtained spectra, the background was subtracted and the position of the peaks determined. The known diffraction angles and intensities allow matching and fitting of the obtained data with the expected patterns. For the surface morphology and cross-section layer structure analysis, Scanning Electron Microscopy (SEM; Vega Tescan, Brno, Czech Republic) was used.

## 2.4. Wetting Surfaces Analysis

For the surfaces wetting analysis, samples were grinded by the papers (up to 400 grit) and rinsed ultrasonically in acetone. The contact angle (CA) of the surfaces was recorded by the optical system with a digital camera (Kruss-DSA25, KRÜSS GmbH, Hamburg, Germany) and measured by dedicated software (Kruss-Advanced 1.5, KRÜSS GmbH, Hamburg, Germany). The surface static CA measurements were conducted on the prepared surfaces with glycerol testing fluid (99.9%, Chemland, Stargard, Poland) and were determined from the geometrical shape of the droplets using the Young-Laplace function and manual baseline correction. The glycerol droplet placement in the sessile drop mode was realized by a special micropipette system with a constant volume of the test liquid (1.5 µL). The data for the analysis were collected with a frequency of 20 fps after droplet placement within duration of 10 s under ambient conditions. The procedure was repeated three times. The selection of the testing fluid was based on preliminary studies that show difficulties in polar and dispersive liquid selection for the initial surface energy calculation assumption. The analyzed pair of water with Diiodomethane or Bromonaphthalene were not available after the HT treatment of the sample surfaces, which rendered the comparison impossible.

## 2.5. Corrosion Resistance Analysis

For the sample corrosion resistance analysis, the potentiodynamic method was used. The experiments and analysis were conducted on Solartron 1285 potentiostat with dedicated (CorrWare, V2.1, Solatron Metrology, West Susex, UK; CorrView, V2.1, Solatron Metrology, West Susex, UK) software. The tests were run in a corrosion cell with two graphite counter electrodes: the SCE (Saturated calomel electrode) reference electrode and the specimen working electrode. For the measurements, the samples were immersed in the Ringer's solution (NaCl: 9 g/L, KCl: 0.42 g/L, CaCl$_2$: 0.48 g/L, NaHCO$_3$: 0.2 g/L) at ambient conditions (23 °C) until the potential change was negligible (1 h). Then, in the assumed potential range (−2.5 to 1 V) with a step size of 1 mV/s, the changes of the current densities were analyzed. For the estimation of the corrosion current densities ($I_{corr}$) and the corrosion potentials ($E_{corr}$), the Tafel extrapolations method for the obtained plots was used. It should be mentioned that in some of the examined cases, the Tafel extrapolation was extremely difficult at the anodic slope range because of the amount of measurement points that represents the linear region. For all examined examples, however, the slope ($\beta_c$) and ($\beta_a$) parameters were reviled for higher clearance. Cathodic ($\beta_c$) and anodic ($\beta_a$) Tafel slopes as well as the corresponding corrosion rates ($P_i$) of the analyzed specimens were extracted from the polarization curves. The corrosion rate ($P_i$), maintains its relation with the corrosion current ($I_{corr}$) in accordance with the equation below [27]:

$$P_i = 22.85 I_{corr} \tag{1}$$

The polarization resistance can also be calculated according to the following equation [27,28]:

$$R_p = (\beta_a \, \beta_c)/(2.3(\beta_a + \beta_c)I_{corr}) \tag{2}$$

### 2.6. Microhardness Measurement Analysis

For the microhardness measurements, the Innovatest Nexus 4000 Vickers tester equipment (INNOVATEST Europe BV, Maastricht, The Netherlands) was used. The hardness of the polished surface of the alloys and their composites was analyzed at a load of 300 g for 10 s with 10 indents per sample to get an average value of the microhardness.

## 3. Results and Discussion

### 3.1. Structural and Morphological Surface Analysis

The structural data, as shown in Figure 2, confirms the HAp layer formation for each sample after the HT process, which is also visible on the SEM images in Figure 3 on the surface and in Figure 4 in the cross-section sample profile view.

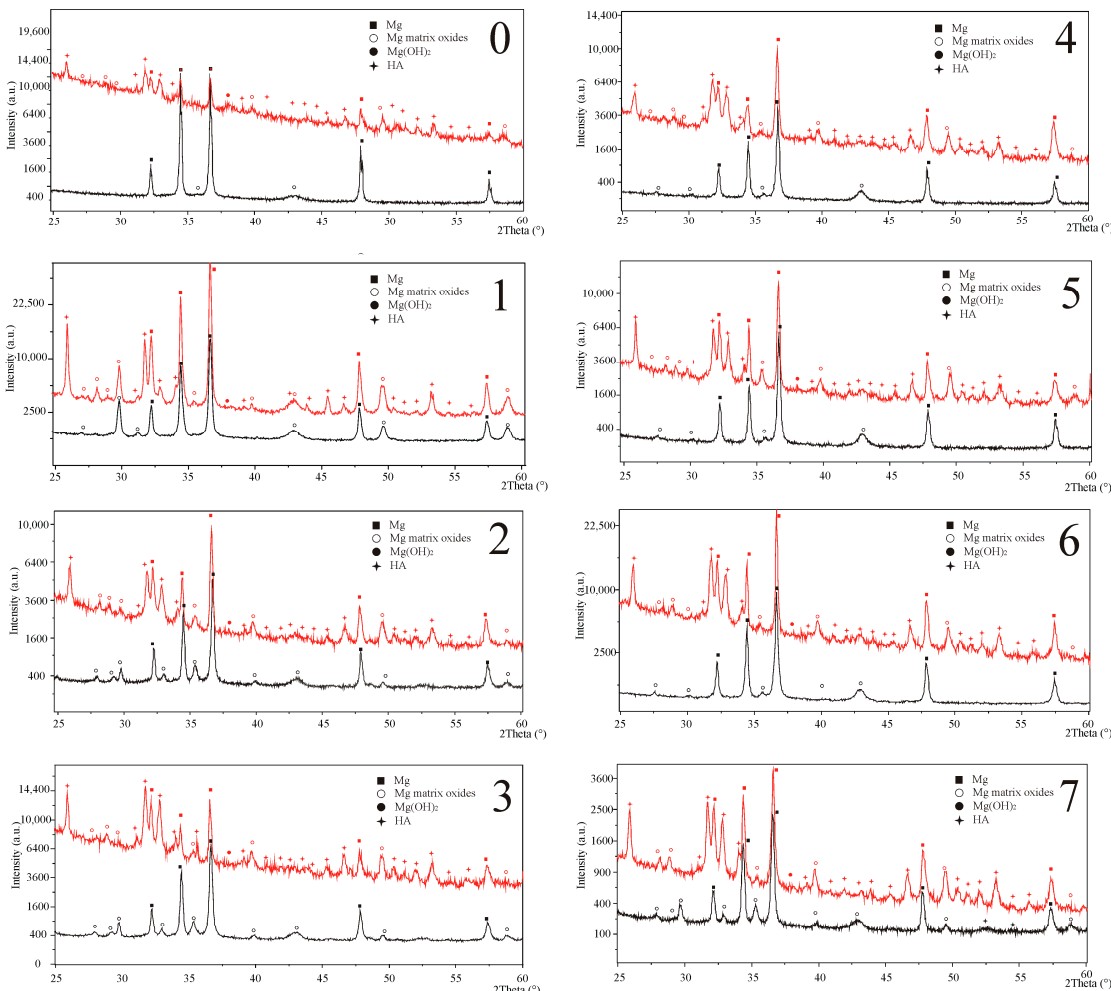

**Figure 2.** XRD spectra of the base (black) and hydrothermally treated (red) samples with detected phases marked in accordance to sample composition. (**0**): Mg pure—Reference sample; (**1**): Mg4Y5.5Dy0.5Zr; (**2**): Mg4Y5.5Dy0.5Zr + 5%BG; (**3**): Mg4Y5.5Dy0.5Zr + 5%BG + 1%Ag; (**4**): Mg1Zn1Mn0.3Zr; (**5**): Mg1Zn1Mn0.3Zr + 5%HA; (**6**): Mg1Zn1Mn0.3Zr + 5%HA + 1%Ag; (**7**): Mg1Zn1Mn0.3Zr + 10%HA.

Beside the hexagonal magnesium (ICDD: 01-035-0821) reflexes, the base sample XRD patterns additionally confirmed the occurrence of oxides (01-071-6485 and 03-065-9075). The analyzed oxides detection level, reliant on the starting composition, may suggest different response of the samples to the HT process environment. The Bioglass reflexes were not observed in the obtained spectra, probably

due to its starting amorphous form and small amounts of the addition that were not detected in our previous studies either [21]:

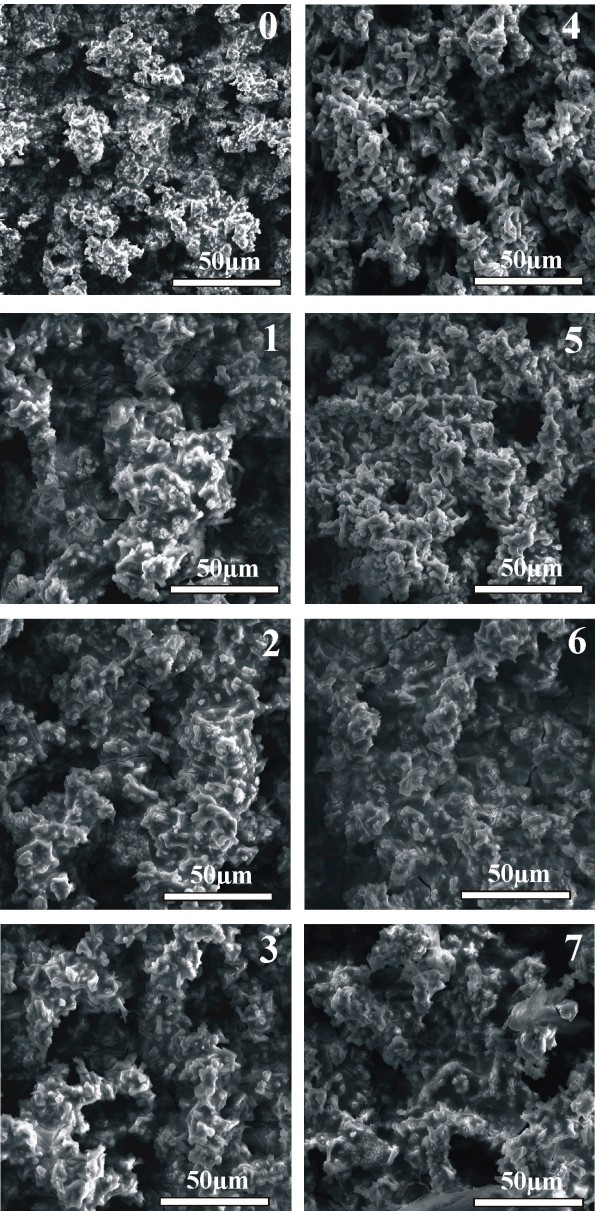

**Figure 3.** Surface view of the obtained hydrothermally treated hydroxyapatite surface layers on the substrates marked in accordance to sample composition. (**0**): Mg pure—Reference sample; (**1**): Mg4Y5.5Dy0.5Zr; (**2**): Mg4Y5.5Dy0.5Zr + 5%BG; (**3**): Mg4Y5.5Dy0.5Zr + 5%BG + 1%Ag; (**4**): Mg1Zn1Mn0.3Zr; (**5**): Mg1Zn1Mn0.3Zr + 5%HA; (**6**): Mg1Zn1Mn0.3Zr + 5%HA + 1%Ag; (**7**): Mg1Zn1Mn0.3Zr + 10%HA.

The hydrothermally treated surface layers exhibit a negligible difference in their composition arising from the substrate. The structurally confirmed occurrence of magnesium hydroxide (01-084-2164) in the modified surfaces remains in accordance with the model of the hydroxyapatite layer formation proposed by other researchers [6,25]. The hydroxyapatite (01-086-1203) layers obtained on the proposed magnesium composites substrates are characterized by homogenous apatite crystal growth. The thickness of the obtained consistent layers shown in Figure 4 varies from 15 to 30 μm. The proposed conditions of the treatment influence the substrate reaction by the visible response of the agent infiltration into the deeper sample section as a changeable spongy conformation (craters and

increased roughness). The morphological view of the layers (Figure 3) confirms their similarity that indicates a stable course of the process on all modified surfaces.

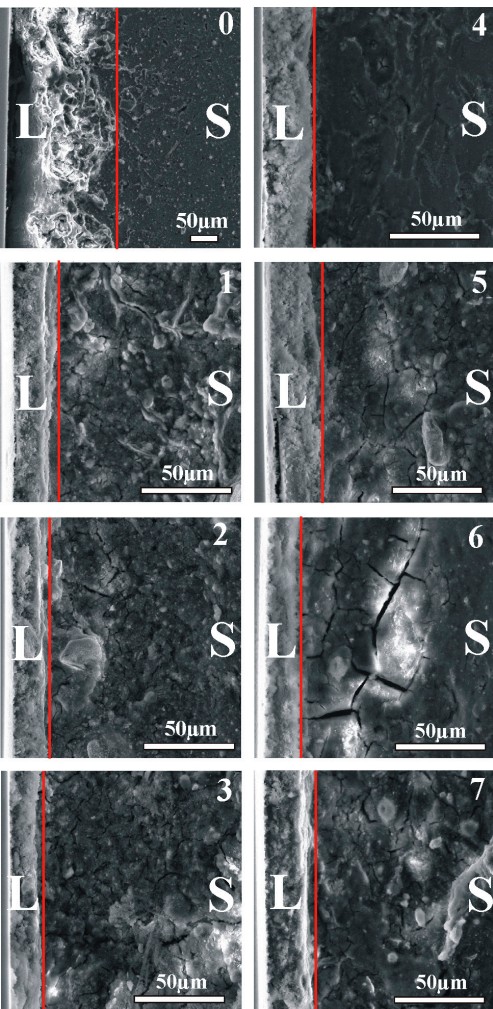

**Figure 4.** Cross-section view of the obtained hydrothermally treated hydroxyapatite surface layers (L) separated by a red line from the substrate (S) material marked in accordance to sample composition. (**0**): Mg pure—Reference sample; (**1**): Mg4Y5.5Dy0.5Zr; (**2**): Mg4Y5.5Dy0.5Zr + 5%BG; (**3**): Mg4Y5.5Dy0.5Zr + 5%BG + 1%Ag; (**4**): Mg1Zn1Mn0.3Zr; (**5**): Mg1Zn1Mn0.3Zr + 5%HA; (**6**): Mg1Zn1Mn0.3Zr + 5%HA + 1%Ag; (**7**): Mg1Zn1Mn0.3Zr + 10%HA.

### 3.2. Surface Wetting Analysis

The glycerol contact angle measurements of the base and the modified surfaces have been shown in Figure 5.

The obtained data confirmed good surface wetting properties after the hydrothermal treatment process. The analyzed contact angle variability as a function of time confirms its impact on the obtained bioactive hydroxyapatite surface layers and on their roughness compared to the base sample results. Lower contact angle results for the hydrothermally treated surfaces (summarized in Table 1) for all the examined examples could be simultaneously confirmed for their average values as well as variability as a function of time. What was also observed for both magnesium composites was that the bioceramic addition to the base alloys and the hydrothermal treatment increase the values of the contact angle. The above relation must originate in the compositional change where, for the composite structures, small additions of oxides decrease the wettability characteristics. The opposite and, at the same time, counterbalance reaction could be observed with a small addition of silver to the main composition of

the proposed complexes. The increase in the surface wettability, associated with the incorporation of silver, also observed in [29], together with other advantages described in the literature [30], form a proper comparative point. The changes of the contact angle as a function of time for the highly viscous glycerol testing fluid, prove the obtained enhancement of the surface bioactivity after the hydrothermal treatment that drastically improves the surface wetting characteristics.

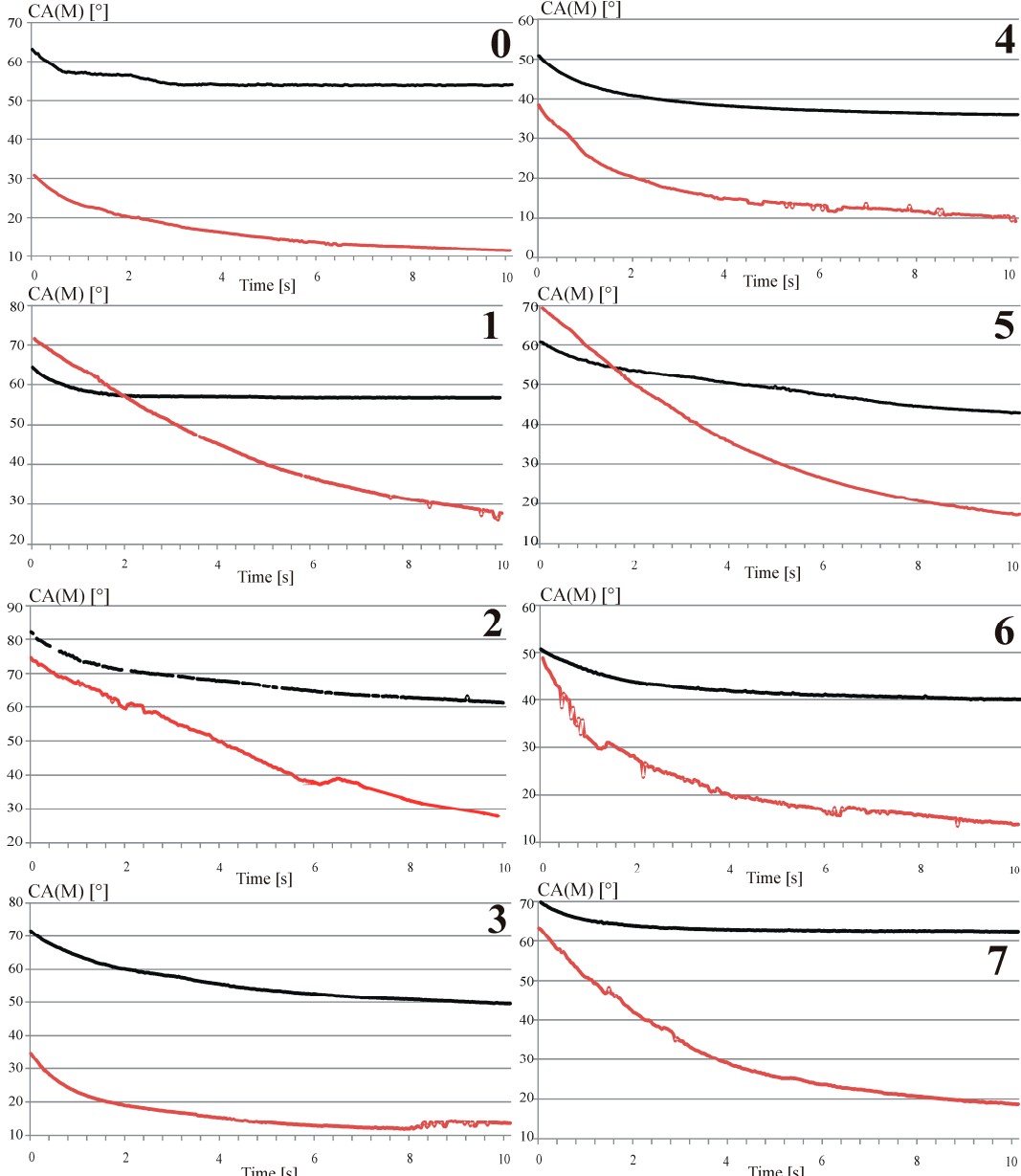

**Figure 5.** The contact angle characteristic as a function of time for the base (black) and hydrothermally treated surfaces (red) marked in accordance to sample composition. (**0**): Mg pure—Reference sample; (**1**): Mg4Y5.5Dy0.5Zr; (**2**): Mg4Y5.5Dy0.5Zr + 5%BG; (**3**): Mg4Y5.5Dy0.5Zr + 5%BG + 1%Ag; (**4**): Mg1Zn1Mn0.3Zr; (**5**): Mg1Zn1Mn0.3Zr + 5%HA; (**6**): Mg1Zn1Mn0.3Zr + 5%HA + 1%Ag; (**7**): Mg1Zn1Mn0.3Zr + 10%HA.

### 3.3. Surface Corrosion Resistance Analysis

The obtained potentiodynamic test results summarized in Figure 6 of the analyzed hydrothermally treated surfaces show two different behaviors for the proposed measurement conditions (compared to the base analogue).

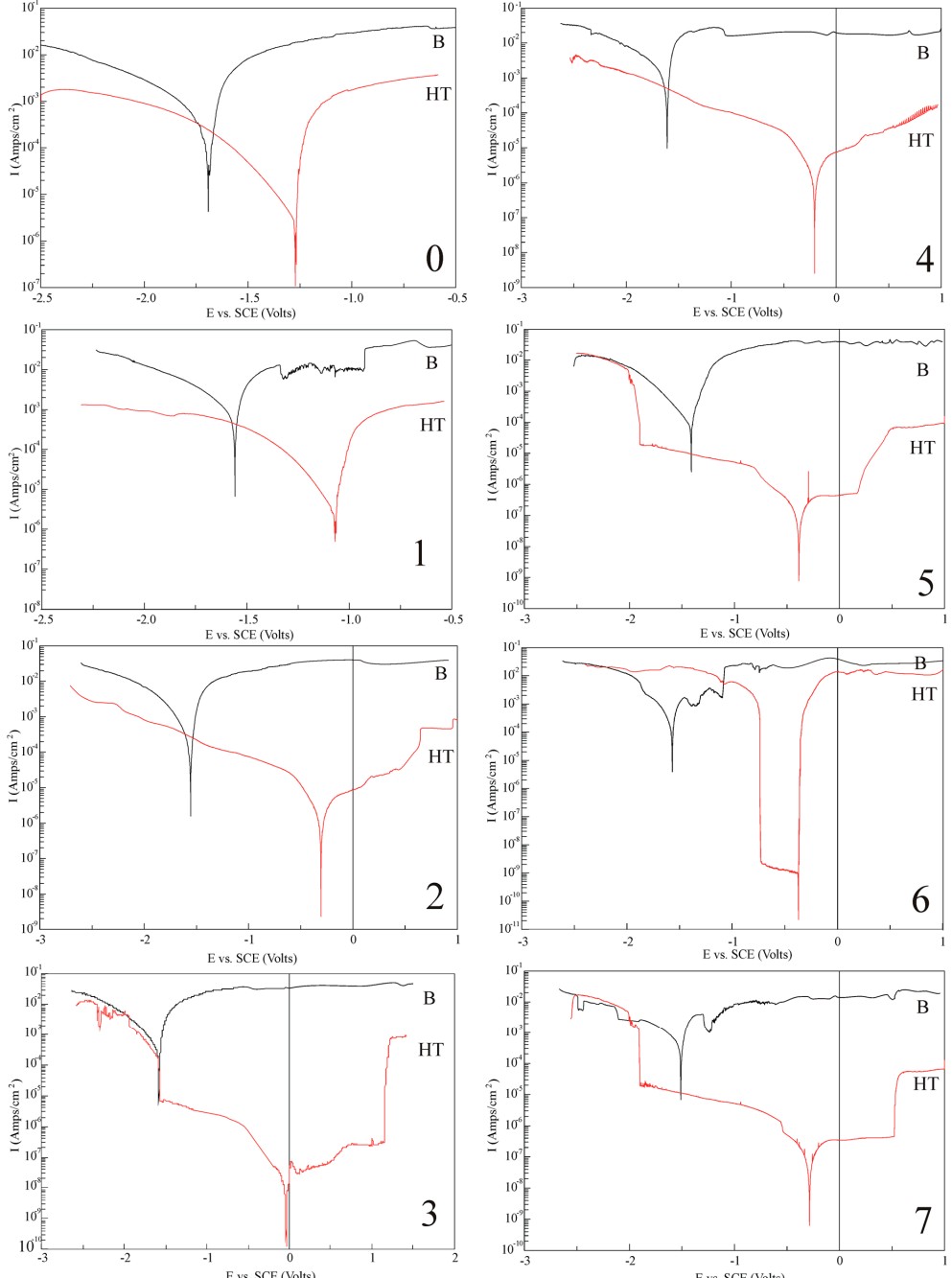

**Figure 6.** Corrosion resistance data based on potentiodynamic curves of analyzed base (black) and hydrothermally treated surfaces (red) marked in accordance to sample composition. (**0**): Mg pure—Reference sample; (**1**): Mg4Y5.5Dy0.5Zr; (**2**): Mg4Y5.5Dy0.5Zr + 5%BG; (**3**): Mg4Y5.5Dy0.5Zr + 5%BG + 1%Ag; (**4**): Mg1Zn1Mn0.3Zr; (**5**): Mg1Zn1Mn0.3Zr + 5%HA; (**6**): Mg1Zn1Mn0.3Zr + 5%HA + 1%Ag; (**7**): Mg1Zn1Mn0.3Zr + 10%HA.

First, the observable relation moves the potential of the modified hydroxyapatite layer to more noble values. Secondly, the corrosion current as well as the corrosion rates of all the analyzed samples after hydrothermal treatment become drastically reduced. Both effects confirm the relevance of the proposed surface treatment and its meaning for the complex composition material with the submicrometer range microstructure as a modification method. High polarization resistance value, calculated from the Equation (2), also implies high corrosion resistance. What was observed in the analysis and confirmed in the summary data in Table 1 is the influence of the addition of silver.

**Table 1.** Summarised properties of the base and hydrothermally treated (HT) samples.

| Sample | Glycerol CA (°) | $I_{corr}$ (µA/cm$^2$) | $E_{corr}$ (V) vs. SCE | Cathodic Slope, $\beta_c$ (mV/decade) vs. SCE | Anodic slope, $\beta_a$ (mV/decade) vs. SCE | Polarization Resistance, RP (kΩ cm$^2$) | Corrosion Rate, $P_i$ (mm/year) | HV0,3 |
|---|---|---|---|---|---|---|---|---|
| 0 | 54.85 (± 1.78) | 230.44 | −1.686 | 208.39 | 90.233 | 0.1188 | 5.265554 | 50 ± 2 |
| 0 HT | 16.34 (± 4.64) | 6.592 | −1.283 | −205.03 | 35.357 | 2.8179 | 0.150627 | - |
| 1 | 43.56 (± 12.86) | 1041.5 | −1.619 | −330.68 | 93.688 | 0.0546 | 23.798275 | 88 ± 2 |
| 1 HT | 32.77 (± 4.77) | 2.49 | −0.870 | −260.44 | 79.45 | 19.9627 | 0.056897 | - |
| 2 | 67.00 (± 4.79) | 484.33 | −1.584 | −292.62 | 92 | 0.1205 | 11.066941 | 95 ± 3 |
| 2 HT | 50.90 (± 12.50) | 1.027 | −0.305 | −194.05 | 98.23 | 84.2177 | 0.023467 | - |
| 3 | 55.52 (± 5.44) | 347.81 | −1.594 | −242.65 | 85.236 | 0.1642 | 7.947459 | 103 ± 2 |
| 3 HT | 16.14 (± 4.78) | 0.00843 | −0.035 | −391.51 | 39.559 | 2269.6064 | 0.000193 | - |
| 4 | 38.86 (± 3.23) | 548.91 | −1.590 | −277.93 | 73.201 | 0.07871 | 12.542594 | 89 ± 2 |
| 4 HT | 16.24 (± 6.48) | 1.968 | −0.195 | −118.56 | 28.421 | 8.2587 | 0.044969 | - |
| 5 | 49.29 (± 4.65) | 328.88 | −1.542 | −325.97 | 99.116 | 0.1883 | 7.514908 | 100 ± 2 |
| 5 HT | 34.66 (± 15.14) | 0.1005 | −0.400 | −180.57 | 160.55 | 6264.6662 | 0.002296 | - |
| 6 | 42.26 (± 2.50) | 239.65 | −1.572 | −119.66 | 66.217 | 0.2689 | 5.476003 | 92 ± 2 |
| 6 HT | 21.26 (± 7.92) | 0.000899 | −0.378 | −533.8 | 15.458 | 7698.8870 | 0.000021 | - |
| 7 | 63.37 (± 1.44) | 452.260 | −1.510 | −440.77 | 60.215 | 0.0671 | 10.334141 | 150 ± 4 |
| 7 HT | 30.74 (± 12.26) | 0.0645 | −0.286 | 125.31 | 143.1 | 450.3386 | 0.001474 | - |

A slight amount of the dopant efficiently decreases the current densities simultaneously for the base and the more accurate HT surfaces. Visible changes in the analyzed current densities could also be observed for the proposed complex composite composition. The bioceramic addition considered for the composite structure preparation enhanced the current characteristics for the base and the more evident HT surfaces. For the composite HT surfaces, a pitting corrosion could also be observed. The above behavior indicates that the obtained HAp layers based on the magnesium hydroxide substrate could be characterized by a different stability in the electrolyte environment. As assumed, based on the conducted research, that the obtained HAp layers possess the same structural conformation, the difference between them may appear at the interlayer magnesium hydroxide subsection. Identical time (for all the samples) of hydrothermal treatment, in the base compositional relation, exhibits a difference in the stability of the samples. For more complex composite substrates, the first stage of the HAp layer formation, the magnesium hydroxide interlayer growth, as well as its tightness may differentiate these substrates significantly.

### 3.4. Analysis of the Microhardness Measurements

The sample composition proposed in this work that allows an obtainment of alloys and their composite structures significantly differs in the measurements of the microhardness. The analysis confirmed that the growing amount of different bioceramic additions to different alloy compositions increases the microhardness results obtained for all the examples. The highest measured value of $150HV_{0.3}$ exceeds the starting values three times in the case of sample 7 (Mg-pure reference sample), in which a significant amount of bioceramic addition (10 wt.% of HA) was applied. The obtained results are characterized with a very low variability, which could also confirm the homogeneous and highly dispersive microstructure.

## 4. Conclusions

The performed research indicates that the analyzed alloys and their composite structures may offer more than just a high dispersion and homogeneous microstructure that allow controlling of the results of the hardness measurements. The proposed HT procedure and its results obtained for the magnesium alloys and the composites that are based on them (with a bioceramic and silver addition) show an evident enhancement in the analyzed properties. The conducted research allows an obtainment of a homogenous and reproducible hydroxyapatite layer on the proposed substrates. The ingrowth HAp layer structure analyzed in this work is composed of apatite crystals of high crystallinity and oxides that may differ in all the examined examples in terms of the relation of the substrate composition. The results obtained for the HT surfaces confirm that the substrate composition may significantly influence the wetting and the corrosion resistance characteristics. Attention was drawn to the possible electrolyte infiltration into the deeper section of the substrate during processing as well as the simultaneously obtained oxide and hydroxide compounds in the layer structure. The obtained results provide strong evidence for the potential application of the HAp layer on the magnesium structures as a corrosion reducer as well as on the bioactive membrane with developed surface morphology for enhanced wetting properties. The combined tackling of the relations in this paper indicates an enhancement of the properties for the HAp layers and the addition of silver.

**Author Contributions:** A.M., K.K., and M.J. conducted the experimental and analytical works as well as wrote the manuscript, M.J. supervised the project. All the authors contributed to the critical reading and editing of the final version of the manuscript.

**Funding:** The research was financially supported by Polish National Science Center 2013/11/B/ST8/04394.

**Conflicts of Interest:** The authors declare no conflict of interest.

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
