# Peer review of "Hydrothermal Surface Treatment of Biodegradable Mg-Materials"

_metals, doi:10.3390/met8110894_

Round 1
Reviewer 1 Report
The present work involve the study of surface treatements on biodegradable magnesium alloys. The work is interesting and the results comprehensive and complete. Hovewer before publication some majior changes have to be performed. In detail:
1) The abstract is not clear and not well written. I suggest to revrite it totally
2) In Fig.2 the labels on XRD spectra are too little and non-readable. I suggest to change it with bigger ones
3) In Fig.4 is not clear what is the substrate and what is the surface layer. I suggest to put some lines and labels in order to identify them.
4) In Fig.5 and Fig.6 the numers on the X-Y axis are too little and non-readable please change it
5) In the whole manuscript the english is not clear, there are a lot of grammatical errors (dots inside the senteces etc) and a lot of sentences without sense. Please check the whole manuscript carefully
Author Response
Dear Reviewer,
We appreciate the opportunity to resubmit our manuscript entitled " "Hydrothermal surface treatment of biodegradable Mg-materials”, to be considered for publication in Metals.
Thank you very much for your comments and suggestions that help considerably to improve the manuscript. We have made major revision to the submitted article (changes are marked in red in the revised manuscript), with answers to the comments point by point below. Moreover, the English grammar and sentence structure were checked and corrected by professional translating company – the changes are marked also by red in the manuscript.
Responses and revision notes: - see att file

Reviewer 2 Report
The paper is written clearly and presents new work. The subject is of high technological and scientific interest. The figures and tables are of enough quality.
1) Corrosion Analysis section. This referee is not convinced with the electrochemical measurements of the corrosion resistance. Time of exposure used in this study (1 hour) seems to be a very short period to asses the changes in electrochemical behavior with the hydrothermal treatment applied for hydroxyapatite layer formation on on corrosion resistance. Due to the corrosion rate may change significantly with time of exposure, I strongly recommend to the authors to use electrochemical measurements (Electrochemical Impedance spectroscopy (EIS)) over extended immersion tests (for example 5 days) to measure the evolution of the protective properties of the specimens. Also, a comparison between the changes in electrochemical data obtained from EIS spectra with time and Hydrogen evolution measurements through 5 days test may be useful for the reader.
2) In Fig 6, it is quite difficult to see how you have put Tafel slopes in. The anodic lines particularly are curved, with no linear region at all. At the very least, the matter should be raised in the manuscript and its accuracy discussed.
Author Response
Dear Reviewer,
We appreciate the opportunity to resubmit our manuscript entitled " "Hydrothermal surface treatment of biodegradable Mg-materials”, to be considered for publication in Metals.
Thank you very much for your comments and suggestions that help considerably to improve the manuscript. We have made major revision to the submitted article (changes are marked in red in the revised manuscript), with answers to the comments point by point below. Moreover, the English grammar and sentence structure were checked and corrected by professional translating company – the changes are marked also by red in the manuscript.
Responses and revision notes: - see att file.

Reviewer 3 Report
Background knowledge about the surface treatment techniques not provided in introduction. The author introduces only Mg alloy and role of each element.
91: the sentence is not completed
99: the image is too small and low resolution
141: stabilization -1h? the sentence is not clear
160: very small images, text in X axis is not clear
How author explains the better properties which are provided by HT, any mechanism or evidence to prove?
240: the first sentence in conclusion is wrong
248-249: Author mentioned maybe substrate composition has a role in obtained properties, need to be evaluated the obtained results come from HT treatment or due to different chemical composition difference?
Mg alloy is very active in aqueous solution and its forming a lot of gas during immersion test which interfere the measurement. Some times it takes forever to reach to the stable potential to perform electrochemical test. Using polarization curve to drive corrosion rate is meaningless if you could not reach to the stable potential. How you confirm the measurements are correct and reproducible?
Author Response
Dear Reviewer,
We appreciate the opportunity to resubmit our manuscript entitled " "Hydrothermal surface treatment of biodegradable Mg-materials”, to be considered for publication in Metals.
Thank you very much for your comments and suggestions that help considerably to improve the manuscript. We have made major revision to the submitted article (changes are marked in red in the revised manuscript), with answers to the comments point by point below. Moreover, the English grammar and sentence structure were checked and corrected by professional translating company – the changes are marked also by red in the manuscript.
Responses and revision notes - see att file.

Round 2
Reviewer 1 Report
Considering that the authors have answered to the major issues revealed in the review process i suggest that the work can be published in the present form
Author Response
Dear Reviewer,
We appreciate the opportunity to resubmit our manuscript entitled " "Hydrothermal surface treatment of biodegradable Mg-materials”, to be considered for publication in Metals.
Thank you very much for your comments and suggestions that help considerably to improve the manuscript. We have made minor revision to the submitted article (changes are marked in red in the revised manuscript), with answers to the comments point by point below.
Responses and revision notes:
Reviewer 1
Considering that the authors have answered to the major issues revealed in the review process i suggest that the work can be published in the present form.
Response - Thank you.

Reviewer 2 Report
Minor points
1) Page 4 line 162 Please change
Pi=28.85Icorr (1)
by
Pi=22.85Icorr (1)
2) Page 4 line 164 Please change
Rp=(βa βc)/(2.3(βa βc)Icorr ) (2)
by
Rp=(βa βc)/(2.3(βa +βc)Icorr ) (2)
Author Response
Dear Reviewer,
We appreciate the opportunity to resubmit our manuscript entitled " "Hydrothermal surface treatment of biodegradable Mg-materials”, to be considered for publication in Metals.
Thank you very much for your comments and suggestions that help considerably to improve the manuscript. We have made minor revision to the submitted article (changes are marked in red in the revised manuscript), with answers to the comments point by point below.
Responses and revision notes:
Reviewer 2
Minor points
1) Page 4 line 164 Please change
Rp=(βa βc)/(2.3(βa βc)Icorr ) (2)
by
Rp=(βa βc)/(2.3(βa +βc)Icorr ) (2)
Response:
Thank you for the comments. All formulas are corrected the final version of the manuscript.

Reviewer 3 Report
Thank you to author for modification of manuscript. However, there are some minor mistakes in the file with possibility of the improvement in scientific parts.
Fig .1: the resolution increased however it is not possible to read the text. Please increase the font size and make a bigger picture and keep it consistent. the font size, and image size differs a lot for one figure.
148, the software called CorrWare and not CorrWear.
224: juxtaposed? please correct the sentence
Author Response
Dear Reviewer,
We appreciate the opportunity to resubmit our manuscript entitled " "Hydrothermal surface treatment of biodegradable Mg-materials”, to be considered for publication in Metals.
Thank you very much for your comments and suggestions that help considerably to improve the manuscript. We have made minor revision to the submitted article (changes are marked in red in the revised manuscript), with answers to the comments point by point below.
Responses and revision notes:
Reviewer 3
Thank you to author for modification of manuscript. However, there are some minor mistakes in the file with possibility of the improvement in scientific parts.
Fig .1: the resolution increased however it is not possible to read the text. Please increase the font size and make a bigger picture and keep it consistent. the font size, and image size differs a lot for one figure.
Thank you. The font size in the Fig was corrected according to reviewer suggestion . The picture is bigger.
148, the software called CorrWare and not CorrWear.
Thank you, The software is called “CorrWare”. The text was corrected.
224: juxtaposed? please correct the sentence
“ The obtained potentiodynamic test results juxtaposed in Fig. 6. “
The sentence was corrected – “The obtained potentiodynamic test results summarized in Fig. 6. “ – see lime 224. Thank you.
